# MetaVoxel: Joint Diffusion Modeling of Imaging and Clinical Metadata

**Yihao Liu**[1]                                                    YIHAO.LIU@VANDERBILT.EDU

**Chenyu Gao**[1]                                                CHENYU.GAO@VANDERBILT.EDU

**Lianrui Zuo**[1]                                                LIANRUI.ZUO@VANDERBILT.EDU

**Michael E. Kim**[2]                                         MICHAEL.KIM@VANDERBILT.DDU

**Brian D. Boyd**[3]                                           BRIAN.D.BOYD@VUMC.ORG

**Lisa L. Barnes**[4]                                          LBARNES1@RUSH.EDU

**Walter A. Kukull**[5]                                       KUKULL@WUSTL.EDU

**Lori L. Beason-Held**[6]                                 HELDLO@GRC.NIA.NIH.GOV

**Susan M. Resnick**[6]                                    RESNICKS@GRC.NIA.NIH.GOV

**Timothy J. Hohman**[7,8]                              TIMOTHY.J.HOHMAN@VUMC.ORG

**Warren D. Taylor**[3,9]                                  WARREN.D.TAYLOR@VUMC.ORG

**Bennett A. Landman**[1,2,10,11]                     BENNETT.LANDMAN@VANDERBILT.EDU

**for the Alzheimer's Disease Neuroimaging Initiative**[*] **and the BIOCARD Study Team**[†]

[1]*Department of Electrical and Computer Engineering, Vanderbilt University, Nashville, TN, US.*

[2]*Department of Computer Science, Vanderbilt University, Nashville, TN, US.*

[3]*Center for Cognitive Medicine, Department of Psychiatry and Behavioral Science, Vanderbilt University Medical Center, Nashville, TN, US.*

[4]*Department of Neurological Sciences and Rush Alzheimer's Disease Center, Rush University Medical Center, Chicago, IL.*

[5]*Washington University in St. Louis, St Louis, MO, US.*

[6]*Laboratory of Behavioral Neuroscience, National Institute on Aging, National Institutes of Health, Baltimore, MD.*

[7]*Vanderbilt Memory and Alzheimer's Center, Vanderbilt University Medical Center, Nashville, TN.*

[8]*Department of Neurology, Vanderbilt University Medical Center, Nashville, Tennessee 37240, USA*

[9]*Geriatric Research, Education, and Clinical Center, Veterans Affairs Tennessee Valley Health System, Nashville, TN, US.*

[10]*Department of Biomedical Engineering, Vanderbilt University, Nashville, TN, US.*

[11]*Department of Radiology, Vanderbilt University Medical Center, Nashville, TN.*

**Editors:** Under Review for MIDL 2026

## Abstract

Modern deep learning methods have achieved impressive results across tasks from disease classification, estimating continuous biomarkers, to generating realistic medical images. Most of these approaches are trained to model conditional distributions defined by a specific predictive direction with a specific set of input variables. We introduce MetaVoxel, a

generative joint diffusion modeling framework that models the joint distribution over imaging data and clinical metadata by learning a single diffusion process spanning all variables. By capturing the joint distribution, MetaVoxel unifies tasks that traditionally require separate conditional models and supports flexible zero-shot inference using arbitrary subsets of inputs without task-specific retraining. Using more than $10,000$ T1-weighted MRI scans paired with clinical metadata from nine datasets, we show that a single MetaVoxel model can perform image generation, age estimation, and sex prediction, achieving performance comparable to established task-specific baselines. Additional experiments highlight its capabilities for flexible inference. Together, these findings demonstrate that joint multimodal diffusion offers a promising direction for unifying medical AI models and enabling broader clinical applicability.

**Keywords:** Diffusion Model, Joint distribution, Multimodal.

## 1. Introduction

Clinicians are routinely faced with a diverse question set when evaluating patients: Does this individual show signs of a particular disease? What is the patient's risk factor given their age? How might this patient's imaging look in two years? How would the imaging appear if disease status were different? These questions highlight the multidimensional nature of clinical assessment, where imaging and clinical metadata together form a complex and interdependent portrait of health.

Deep learning have provided the tools to help address these questions. Modern classifiers can accurately predict disease status from imaging (Li et al., 2015; Paul et al., 2017). Regression models can estimate continuous attributes such as age or risk scores with impressive precision (Venkadesh et al., 2021; Gao et al., 2025). Generative models can synthesize realistic medical images for data augmentation (Guo et al., 2025; Zhang et al., 2025), and predict disease trajectories (Puglisi et al., 2025). While each of these task-specific methods has advanced considerably, every task requires a separate model with a predefined set of inputs. This contrasts with clinical reasoning, where the boundaries between questions are fluid, and reasoning often shifts from diagnosis to prognosis to hypothetical scenarios within the course of a single patient encounter.

In this work, we present MetaVoxel, a generative diffusion framework designed to capture the multidimensional nature of patients. Unlike existing approaches that learn a conditional distribution of a target variable given a predefined set of inputs variables, we model the joint distribution over all variables, including imaging and clinical metadata. Although this is substantially more challenging than focusing on a single predictive direction, joint modeling provides broader opportunities. Because the joint distribution encompasses **all possible**

∗. Data used in preparation of this article were obtained from the Alzheimer's Disease Neuroimaging Initiative (ADNI) database (adni.loni.usc.edu). As such, the investigators within the ADNI contributed to the design and implementation of ADNI and/or provided data but did not participate in analysis or writing of this report. A complete listing of ADNI investigators can be found at: http://adni.loni.usc.edu/wp-content/uploads/how_to_apply/ADNI_Acknowledgement_List.pdf

†. Data used in preparation of this article were derived from BIOCARD study data, supported by grant U19 –AG033655 from the National Institute on Aging. The BIOCARD study team did not participate in the analysis or writing of this report, however, they contributed to the design and implementation of the study. A listing of BIOCARD investigators may be accessed at: https://www.biocard-se.org/public/Core%20Groups.html

**conditionals**, a single trained diffusion model can perform diverse tasks beyond image generation. Moreover, by capturing the complete set of dependencies among variables, our model enables flexible conditioning on arbitrary subsets of information at test time. Our contributions are summarized as follows:

- We introduce MetaVoxel, a multimodal diffusion model that learns the joint distribution over imaging data and clinical metadata;

- We develop a zero-shot inference procedure that allows MetaVoxel to access conditional distributions and perform diverse tasks using arbitrary subsets of input variables.

- Using more than 10,000 paired MRI–metadata samples, we demonstrate that MetaVoxel achieves competitive performance on image generation, age estimation, and sex prediction, while supporting flexible conditioning scenarios.

## 2. Backgrounds and Related Works

**Denoising Diffusion Probabilistic Models (DDPMs)** (Sohl-Dickstein et al., 2015; Ho et al., 2020) are generative models that aim to learn the underlying data distribution $p(x)$ by defining a forward diffusion process that gradually corrupts data, and a reverse process that learns to invert this corruption process. The forward process is a fixed Markov chain that incrementally adds noise to a data sample $x_0$ over a sequence of discrete time steps $t = 1, \ldots, T$. In practice, Gaussian noise is the most common choice for image generation tasks, where the forward process provides a smooth trajectory from data $x_0$ to a latent variable $x_T$ that follows a simple prior distribution. Each successive latent variable $x_t$ along this trajectory follows

$$q(x_t|x_{t-1}) = \mathcal{N}\left(x_t \mid \sqrt{1 - \beta_t^{\mathrm{DDPM}}}\, x_{t-1},\, \beta_t^{\mathrm{DDPM}} I\right), \tag{1}$$

where $\beta_t^{\mathrm{DDPM}} \in (0, 1]$ controls the noise magnitude at each step. The reverse process, parameterized by a time-conditioned neural network, learns to denoise step-by-step by predicting the mean of the conditional distribution $p_\theta(x_{t-1}|x_t)$. The model is trained by maximizing a variational lower bound on the data likelihood. In practice, this objective simplifies to mean squared error terms between the true and predicted total noise added to an image $x_t$ across all time steps $t$. Once trained, new samples can be generated by drawing an initial latent variable $x_T$ from a standard Gaussian distribution and iteratively applying the learned reverse process to obtain a realistic data sample.

Several extensions have been proposed to improve the efficiency and scalability of DDPMs. Denoising Diffusion Implicit Models (DDIMs) (Song et al.) introduce an implicit probabilistic formulation that shares the same training objective as DDPMs but enables deterministic sampling that requires significantly fewer denoising steps. Latent Diffusion Models (LDMs) (Rombach et al., 2022) enhance the framework by performing diffusion in a latent space learned via a variational autoencoder, achieving substantial computational efficiency for high-resolution image generation.

**Discrete diffusion models**, such as Discrete Denoising Diffusion Probabilistic Models (D3PMs) (Austin et al., 2021) and (Hoogeboom et al., 2021), generalize the diffusion

process to categorical variables. In this setting, the forward process is defined as

$$q(\mathbf{x}_t|\mathbf{x}_{t-1}) = \text{Cat}(\mathbf{x}_t; \mathbf{p} = \mathbf{x}_{t-1}\mathbf{Q}_t), \tag{2}$$

where $\text{Cat}(\mathbf{x}; \mathbf{p})$ denotes a categorical distribution over the one-hot row vector $\mathbf{x}$ with class probabilities given by the row vector $\mathbf{p}$, and the transition matrices $\mathbf{Q}_t$ control the corruption process. The reverse process typically adopts an $\mathbf{x}_0$-parameterization, where a time-conditioned neural network models $p_\theta(\mathbf{x}_0|\mathbf{x}_t)$. From here, one can derive $p_\theta(\mathbf{x}_{t-1}|\mathbf{x}_t)$, allowing sampling to proceed from an initial random one-hot vector through iterative denoising, analogous to the procedure used in DDPMs.

**Conditional DDPMs** extend the diffusion framework to learn conditional distributions $p(x|y)$, enabling sampling guided by auxiliary information $y$. This formulation has become foundational across many tasks, such as text-to-image and structure-guided image synthesis. Prominent examples include Stable Diffusion (Rombach et al., 2022) and ControlNet (Zhang et al., 2023), which enable flexible and controllable image generation from text prompts or structural inputs. In medical research, conditional diffusion models have been explored to address data scarcity problem in AI model development (Guo et al., 2025) and disease progression modeling (Puglisi et al., 2025; McMaster et al., 2025).

**Unconditioned DDPMs** can also be adapted for **conditional generation** during the sampling process, as demonstrated by methods such as RePaint (Lugmayr et al., 2022). RePaint enables free-form image inpainting, where the goal is to fill in unknown regions of an image specified by an arbitrary binary mask, while maintaining fidelity in the known regions and ensuring global coherence. At each time step $t$ during sampling, the updated sample $x_{t-1}$ is composed by combining the re-noised known region $(x_{t-1}^{\text{known}})$, obtained through the forward process, with the denoised unknown region $(\tilde{x}_{t-1}^{\text{unknown}})$ generated by the reverse process. This process can be written as:

$$\tilde{x}_{t-1} = m \odot x_{t-1}^{\text{known}} + (1 - m) \odot \tilde{x}_{t-1}^{\text{unknown}}, \tag{3}$$

where $m$ is a binary mask indicating the known regions. This composition step, applied at every time step $t$ during sampling, preserves consistency in the known regions while generating coherent and realistic content in the unknown regions.

## 3. Method

Clinical data inherently involve multiple interdependent variables that extend beyond imaging alone. While existing diffusion-based models primarily focus on images, we treat all variables as equally important components. Without loss of generality, we consider three specific variables in this work: T1-weighted (T1w) magnetic resonance (MR) image $I$, age $A$, and sex $S$. These variables were selected because they represent the types of information most commonly available in existing datasets, spanning both high-dimensional imaging data and scalar metadata, encompassing both continuous and categorical data types. MetaVoxel naturally extends to additional variables with minimal modification. The remainder of this section is organized as follows. Section 3.1 introduces the MetaVoxel framework and its formulation for learning the joint distribution over $(I, A, S)$. A visual diagram is provided in Figure 1. Section 3.2 describes how MetaVoxel enables flexible inference using arbitrary subsets of inputs to perform image generation, regression, and classification.

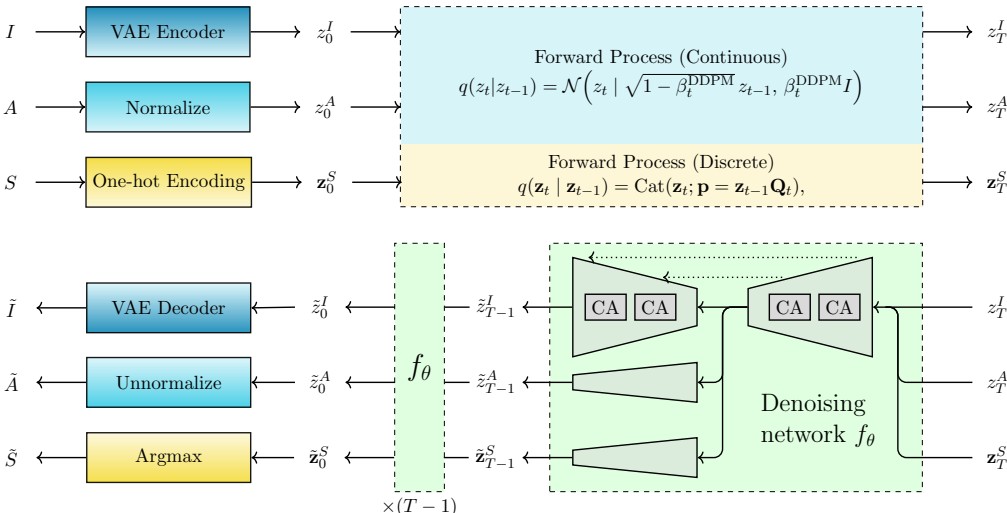

Figure 1: Schematic of the MetaVoxel diffusion framework. Continuous variables such as the image $I$ and age $A$ undergo Gaussian diffusion, while discrete variables such as sex $S$ follow a discrete diffusion process. A single denoising network $f_\theta$ models the shared reverse process.

### 3.1. Learning the Joint Distribution

**Encoding.** To model the joint distribution, MetaVoxel must encode each variable in a form suitable for inclusion in a single joint diffusion process, while allowing each variable to retain an appropriate representation. For the imaging variable $I$, which consists of high-resolution 3D MR volumes, dimensionality reduction is essential to make diffusion modeling computationally feasible. We adopt a KL-regularized variational autoencoder (VAE) in which an encoder network $\mathcal{E}$ maps an image $i \in \mathbb{R}^{H \times W \times D}$ to a compact latent representation $z^I = \mathcal{E}(i)$, $z^I \in \mathbb{R}^{\frac{H}{8} \times \frac{W}{8} \times \frac{D}{8}}$, and a decoder $\mathcal{D}$ reconstructs the image from this latent space. A light KL penalty is applied to keep the distribution of $z^I$ close to a standard normal, which prevents arbitrarily high-variance latent spaces. We implement this VAE by extending the 2D architecture used in LDM (Rombach et al., 2022) to full 3D, and we train it using the same loss formulation employed in MAISI (Guo et al., 2025).

Scalar metadata, in contrast, do not require dimensionality reduction for computational reasons. We apply lightweight encoding process to place them in numerically stable ranges and representations compatible with MetaVoxel's diffusion process. Continuous variables such as age are linearly scaled to lie approximately within the interval $[-1, 1]$, ensuring numerical stability similar to that imposed on the image latent space. Categorical variables such as sex are converted to one-hot vectors. We denote these encoded scalar variables as $z^A$ and $\mathbf{z}^S$, respectively.

**Joint Diffusion.** Given the encoded representations, MetaVoxel applies a forward diffusion process to each variable using a corruption mechanism suited to its representation. For image latents $z^I$ and continuous scalar variables $z^A$, the forward process follows the

standard Gaussian diffusion defined in Equation 1. Categorical variables $\mathbf{z}^S$ follow the discrete corruption process defined in Equation 2. We adopt transition matrix

$$\mathbf{Q}_t = (1 - \beta_t^{\text{D3PM}})\mathbf{I} + \beta_t^{\text{D3PM}}/K\mathbb{1}\mathbb{1}^T, \tag{4}$$

where $K$ is the number of categories, $\beta_t^{\text{D3PM}} \in [0, 1]$ follows a cosine schedule. All variables share the same time index $t$, so each diffusion step produces a single noisy tuple $z_t = (z_t^I, z_t^A, \mathbf{z}_t^S)$.

MetaVoxel learns a unified reverse process that jointly denoises all variables. A single time-conditioned denoising network $f_\theta(z_t, t)$ takes the full noisy tuple $z_t$ as input and predicts the quantities required to reverse their respective corruption processes: Gaussian noise for continuous components and logits for categorical components. During training, the model is optimized using a combined objective that mirrors the simplified evidence lower bound (ELBO) formulations for both Gaussian and discrete diffusion. For the image latent and continuous scalar variables, ELBO collapses into mean-squared error losses between the true and the predicted noise. For categorical variables, the ELBO under the $x_0$-parameterization reduces to a cross-entropy loss between the true class label and the logits predicted by $f_\theta(z_t, t)$. The overall training objective is therefore

$$\mathcal{L} = \mathbb{E}_{z_t, t, \epsilon^I \sim \mathcal{N}(0,\mathbf{I}), \epsilon^A \sim \mathcal{N}(0,\mathbf{I})} \left[ \frac{m^3}{HWD} \left\| \epsilon^I - f_\theta(z_t, t)^I \right\|_2^2 + \left\| \epsilon^A - f_\theta(z_t, t)^A \right\|_2^2 \right]$$
$$+ \mathbb{E}_{z_0, t, z_t \sim q(\cdot|z_0)} \left[ \text{CE} \left( \mathbf{z}_0^S, \text{softmax}(f_\theta(z_t, t)^S) \right) \right], \tag{5}$$

where the superscripts on $f_\theta(z_t, t)$ indicate the component of the network output corresponding to each variable and $\text{CE}(\cdot, \cdot)$ refers to the cross-entropy loss.

The denoising network $f_\theta$ must process all variables jointly and produce outputs aligned with the structure of each variable. To accommodate this multimodal input–output structure, we adapt the U-Net architecture used in LDM into a 3D model. The scalar variables are integrated into the model through a combination of input-channel concatenation and cross-attention (CA). At the input, discrete scalar variables are represented as single scalars rescaled to the interval $[-1, 1]$, consistent with continuous variables. On the output side, the standard U-Net decoder produces the prediction for image variables, while additional lightweight decoding heads are attached to the bottleneck features for the scalar variables (see Figure 1). Each scalar head consists of two (GroupNorm $\rightarrow$ SiLU $\rightarrow$ Conv) blocks with a skip connection, followed by global average pooling to produce a single output value.

### 3.2. Zero-shot Inference with Arbitrary Conditioning

A trained MetaVoxel model can be *unconditionally sampled* by drawing Gaussian or categorical noise for each variable at timestep $T$, and iteratively apply the denoising network $f_\theta$ until reaching $t = 0$. Unlike conventional diffusion models that focus solely on image generation, MetaVoxel can generate coherent synthetic patient profiles from the joint distribution $p(I, A, S)$, as shown in Figure 2.

A distinctive strength of MetaVoxel is its ability to perform flexible zero-shot inference for a broad spectrum of tasks with arbitrary subsets of inputs. To enable this, we reinterpret the RePaint strategy described in Section 2: just as RePaint performs conditional image

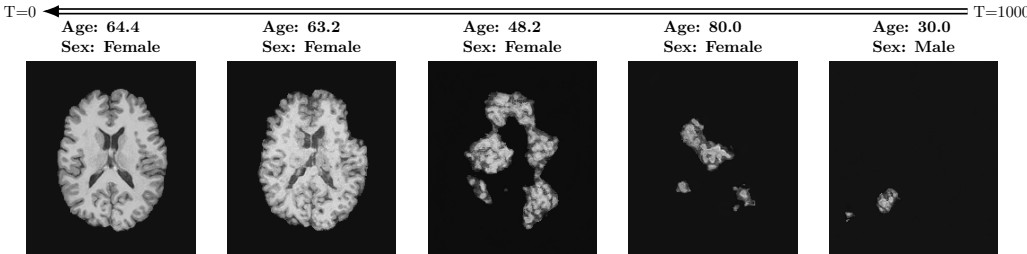

Figure 2: Unconditional sampling in MetaVoxel. At time step $T = 1000$, Gaussian noise is sampled for image latents and continuous variables, and categorical noise for categorical variables. The variables are then jointly denoised from $T = 1000$ to $T = 0$ to generate coherent samples from the learned joint distribution. All images, ages, and sexes shown are decoded from the latent space before visualization. Additional examples can be found in Appendix (Figure B.1).

generation by specifying a binary mask that marks which pixels are fixed, we can further designate any subset of MetaVoxel's variables as "known" conditions. At each denoising step, these known variables are overwritten with their re-noised values, ensuring that they remain **fixed** throughout the sampling trajectory, while the remaining variables evolve according to the learned reverse process. This keeps the synthesized variables consistent with the conditioning, analogous to how RePaint produces inpainted regions consistent with the known region. Although MetaVoxel learns the joint distribution $p(I, A, S)$, different downstream tasks can be realized simply by choosing which variables to fix during sampling. Image inpainting is achieved by fixing pixel regions of the image variable[*]; conditional image generation by fixing variables such as age or sex; regression tasks (*e.g.,* age estimation) by fixing any combination of image and sex; and classification tasks (*e.g.,* sex prediction) by fixing any combination of image and age. As more variables are included in the joint model, the range of possible zero-shot tasks naturally expands. For example, when multiple imaging modalities are present, fixing one modality and sampling the other enables image-to-image translation.

## 4. Experiments

MetaVoxel introduces several architectural and functional components that are rare, and in combination, not seen in existing work. While prior efforts have addressed isolated components, none to our knowledge provide a unified framework with the combined capabilities of MetaVoxel. As a result, direct head-to-head comparisons with a single baseline are not feasible. Given this context, our experiments were structured around three key questions:

1. How does introducing discrete diffusion influence the quality of image generation compared to image-only diffusion baselines?

2. How does MetaVoxel perform on regression versus specialized regression models?

---

*. see examples in Appendix (Figure B.2)

3. How does MetaVoxel perform on classification versus specialized classification models?

**Datasets**: We compiled a cohort of 10,154 T1-weighted (T1w) brain MR scans with age and sex information. These scans were drawn from nine datasets: ADNI (Jack Jr et al., 2008), BIOCARD (Sacktor et al., 2017), BLSA (Shock, 1984), HCPA (Bookheimer et al., 2019; Harms et al., 2018), ICBM (Mazziotta et al., 2001), NACC (Beekly et al., 2007), OASIS-3 (LaMontagne et al., 2019), ROS/MAP/MARS (Bennett et al., 2018; L Barnes et al., 2012), and WRAP (Johnson et al., 2018). Additional information on ADNI and NACC can be found in Appendix A. Only cognitively unimpaired individuals were included. All images underwent standardized preprocessing consisting of N4 inhomogeneity-field correction (Tustison et al., 2010) and skull stripping using HD-BET (Isensee et al., 2019); for subjects with multiple imaging sessions, the baseline scan was first rigidly registered using ANTs (Avants et al., 2009) to an MNI template (Fonov et al., 2009), and all follow-up scans were subsequently rigidly registered to the registered baseline. Final preprocessed images underwent manual quality assurance by the authors using a custom Python-based application interface (Kim et al., 2025). We then performed a **subject-level** data split, allocating approximately 89% of subjects for training, 1% for validation, and 10% for testing, resulting in 9,078 cases for training, 108 for validation, and 968 for testing.

We trained a single MetaVoxel model to jointly model the T1w image, age, and sex using the training set. **Unless otherwise stated, all MetaVoxel results were generated from this single model.** The validation set was used to select the optimal training epoch. To accelerate the sampling process, we used DDIM with 50 steps for continuous variables and k-step sampling ($k = 20$) for discrete variables. All baseline methods used the same data split and the same protocol for model selection and evaluation.

### 4.1. Impact of Discrete Diffusion on Image Generation

Discrete diffusion is essential in incorporating categorical variables, such as sex. However, introducing a discrete diffusion pathway alongside Gaussian diffusion may influence image generation quality. Since joint diffusion model has not been explored in prior work, the effect of combining these two diffusion processes remains unknown. To evaluate how discrete diffusion affects image generation, we compared MetaVoxel to two baselines: (a) LDM: a latent diffusion model trained solely on T1w images, and (b) Continuous-Sex MetaVoxel: a variant of MetaVoxel in which the sex variable is treated as an extra continuous scalar and modeled using Gaussian diffusion.

For each model, we generated 100 synthetic T1w scans and quantified sample quality using the Fréchet Inception Distance (FID). FID measures the distance between the distribution of generated images and the distribution of real images from the held-out test set by comparing their activations in a pretrained network. To account for the 3D nature of the data, we computed FID using the approach implemented in MAISI (Guo et al., 2025): each 3D T1w scan was sliced along the axial, coronal, and sagittal planes, FID was computed separately for each plane using the corresponding slice-level feature distributions, and the three values were averaged to obtain the final FID.

Although LDM converges in roughly one-third the training time required by MetaVoxel and Continuous-Sex MetaVoxel, the three models achieve comparable FID (Table 1), with no degradation observed when adding either continuous or discrete diffusion pathways. Given

Table 1: Quantitative results across image generation, regression, and classification tasks. An em-dash (–) indicates cases where a model, under its specific training setup, is not applicable or cannot perform the corresponding task.

| Method | Image Generation
FID ↓ | Age Estimation
MAE ↓ | Sex Prediction
ACC ↑ |
|---|---|---|---|
| LDM (Rombach et al., 2022) | 10.94 | – | – |
| 3D-DenseNet(MSE) (Huang et al., 2017) | – | **3.96 ± 2.92** | – |
| 3D-ViT(MSE) (Dosovitskiy, 2020) | – | 7.99 ± 6.72 | – |
| BRAID-T1w (Gao et al., 2025) | – | 4.01 ± 3.31 | – |
| 3D-Dense(CE) (Huang et al., 2017) | – | – | **0.884** |
| 3D-ViT(CE) (Dosovitskiy, 2020) | – | – | 0.788 |
| MetaVoxel | **10.84** | 4.50 ± 3.46 | 0.815 |
| Continuous-Sex MetaVoxel | 11.18 | 4.95 ± 4.16 | 0.855 |

that the continuous-sex variant cannot naturally extend beyond binary categories, discrete diffusion offers a practical and scalable way to represent categorical data within the joint model. These results indicate that discrete diffusion can be integrated into a joint generative framework without compromising image sample quality.

## 4.2. Regression and Classification

Although diffusion models have recently matured as powerful image generative methods, there has been little incentive to develop dedicated conditional diffusion models for regression or classification. These tasks already have well-established solutions, and in most settings the diffusion sampling process offers no clear advantage. MetaVoxel sidesteps the need of dedicated conditional models for accessing conditional distributions such as $p(A|I,S)$ or $p(S|I,A)$. To compare MetaVoxel with established approaches, we examine its performance on age estimation (regression) and sex prediction (classification).

**Age estimation**: we sampled MetaVoxel three times with $I$ and $S$ treated as known variables during the sampling process. The predicted age was obtained by averaging the three sampled values. We then reported the mean absolute error between predicted and true age for MetaVoxel and the following comparison methods: (a) 3D-DenseNet(MSE): A 3D DenseNet that receives the T1w image with sex concatenated at the input channel level and is trained using mean squared error (MSE) loss; (b) ViT(MSE): A Vision Transformer that receives the T1w image with sex concatenated at the input channel level and is trained with MSE loss; (c) BRAID-T1w: A ResNet-based architecture that incorporates T1w image and sex information at the feature level and is trained with MSE loss.

**Sex prediction**: we sampled MetaVoxel three times with $I$ and $A$ as known variables in the sampling process. We then used majority voting to determine the prediction and we reported classification accuracy on MetaVoxel and the following methods: (a) 3D-DenseNet(CE): A 3D DenseNet that receives the T1w image with age concatenated at the input channel level and is trained with cross-entropy loss; (b) ViT(CE): A Vision Transformer

Table 2: Flexible conditioning of MetaVoxel. MetaVoxel is evaluated by sampling age under four conditioning settings: (I, S): conditioned on both the T1w MR scan and sex; (I): conditioned only on the T1w MR scan; (S): conditioned only on sex; and ($\varnothing$): the unconditional setting with no fixed variables.

|  | MetaVoxel($I, S$) | MetaVoxel($I$) | MetaVoxel($S$) | MetaVoxel($\varnothing$) | Population Mean |
|---|---|---|---|---|---|
| MAE | $4.50 \pm 3.46$ | $4.57 \pm 3.54$ | $11.34 \pm 9.15$ | $11.27 \pm 8.96$ | $10.76 \pm 7.76$ |
| Mean Sample Variance | 9.76 | 9.88 | 62.89 | 64.61 | – |

that receives the T1w image with age concatenated at the input channel level and is trained with CE loss.

Table 1 shows that MetaVoxel's performance on both age estimation and sex prediction is well within the range of established discriminative models. Its primary drawback is computational cost: generating a single prediction requires roughly 30 seconds on a Nvidia A6000 GPU, whereas the baseline models produce outputs in under a second. Despite this disadvantage, the results demonstrate that a trained MetaVoxel model can be used for regression and classification without any task-specific retraining. Moreover, because MetaVoxel models the entire joint distribution rather than only conditional means (in regression) or decision boundaries (in classification), it naturally supports flexible conditioning scenarios and sample-based uncertainty that are not directly available to conventional discriminative models. To illustrate this, we use age estimation as an example. With the same MetaVoxel model, we generated age samples under any subset of observed variables (*e.g.,* T1w scans, sex, both, or none) thereby accessing $p(A|I, S)$, $p(A|I)$, $p(A|S)$, and $p(A)$ without retraining. For each conditioning choice, we used the average of three samples of $A$ to obtain the predicted age, and compared it with the true age. We summarize the results in Table 2. The MAE depends almost entirely on whether the image is provided. Adding sex as an additional variable offers no measurable benefit, and using only sex or no information yields MAEs comparable to using population mean of the training data as prediction. We also find that, when the image is absent, the mean sample variance increases substantially, reflecting greater uncertainty that closely tracks the rise in MAE.

## 5. Conclusion

We introduced MetaVoxel, a multimodal diffusion framework that learns a single joint generative model over imaging and clinical metadata. By modeling the full joint distribution, MetaVoxel unifies tasks that traditionally require separate conditional architectures. Experiments on more than $10,000$ T1-weighted magnetic resonance scans demonstrate that this single generative model can perform image generation, age estimation, and sex prediction with performance comparable to established baselines, despite relying solely on diffusion-based sampling at inference. Beyond matching task-specific models, MetaVoxel enables zero-shot inference from arbitrary subsets of imaging and metadata. These results highlight the potential of joint multimodal diffusion modeling as a foundation for general-purpose medical AI systems.

## Acknowledgments

Data collection and sharing for this project was funded (in part) by the Alzheimer's Disease Neuroimaging Initiative (ADNI) (National Institutes of Health Grant U01 AG024904) and DOD ADNI (Department of Defense award number W81XWH-12-2-0012). ADNI is funded by the National Institute on Aging, the National Institute of Biomedical Imaging and Bioengineering, and through generous contributions from the following: AbbVie, Alzheimer's Association; Alzheimer's Drug Discovery Foundation; Araclon Biotech; BioClinica, Inc.; Biogen; Bristol-Myers Squibb Company; CereSpir, Inc.; Cogstate; Eisai Inc.; Elan Pharmaceuticals, Inc.; Eli Lilly and Company; EuroImmun; F. Hoffmann-La Roche Ltd and its affiliated company Genentech, Inc.; Fujirebio; GE Healthcare; IXICO Ltd.; Janssen Alzheimer Immunotherapy Research & Development, LLC.; Johnson & Johnson Pharmaceutical Research & Development LLC.; Lumosity; Lundbeck; Merck & Co., Inc.; Meso Scale Diagnostics, LLC.; NeuroRx Research; Neurotrack Technologies; Novartis Pharmaceuticals Corporation; Pfizer Inc.; Piramal Imaging; Servier; Takeda Pharmaceutical Company; and Transition Therapeutics. The Canadian Institutes of Health Research is providing funds to support ADNI clinical sites in Canada. Private sector contributions are facilitated by the Foundation for the National Institutes of Health (www.fnih.org). The grantee organization is the Northern California Institute for Research and Education, and the study is coordinated by the Alzheimer's Therapeutic Research Institute at the University of Southern California. ADNI data are disseminated by the Laboratory for Neuro Imaging at the University of Southern California.

The BLSA is supported by the Intramural Research Program, National Institute on Aging, NIH.

The BIOCARD study is supported by a grant from the National Institute on Aging (NIA): U19-AG03365. The BIOCARD Study consists of 7 Cores and 2 projects with the following members: (1) The Administrative Core (Marilyn Albert, Corinne Pettigrew, Barbara Rodzon); (2) the Clinical Core (Marilyn Albert, Anja Soldan, Rebecca Gottesman, Corinne Pettigrew, Leonie Farrington, Maura Grega, Gay Rudow, Rostislav Brichko, Scott Rudow, Jules Giles, Ned Sacktor); (3) the Imaging Core (Michael Miller, Susumu Mori, Anthony Kolasny, Hanzhang Lu, Kenichi Oishi, Tilak Ratnanather, Peter vanZijl, Laurent Younes); (4) the Biospecimen Core (Abhay Moghekar, Jacqueline Darrow, Alexandria Lewis, Richard O'Brien); (5) the Informatics Core (Roberta Scherer, Ann Ervin, David Shade, Jennifer Jones, Hamadou Coulibaly, Kathy Moser, Courtney Potter); the (6) Biostatistics Core (Mei-Cheng Wang, Yuxin Zhu, Jiangxia Wang); (7) the Neuropathology Core (Juan Troncoso, David Nauen, Olga Pletnikova, Karen Fisher); (8) Project 1 (Paul Worley, Jeremy Walston, Mei-Fang Xiao), and (9) Project 2 (Mei-Cheng Wang, Yifei Sun, Yanxun Xu.

Data collection and sharing for this project was provided by the Human Connectome Project (HCP; PI: Bruce Rosen, M.D., Ph.D., Arthur W. Toga, Ph.D., Van J. Weeden, MD). HCP funding was provided by the National Institute of Dental and Craniofacial Research (NIDCR), the National Institute of Mental Health (NIMH), and the National Institute of Neurological Disorders and Stroke (NINDS). HCP data are disseminated by the Laboratory of Neuro Imaging at the University of Southern California.

Data collection and sharing for this project was provided by the International Consortium for Brain Mapping (ICBM; Principal Investigator: John Mazziotta, MD, PhD). ICBM funding

was provided by the National Institute of Biomedical Imaging and BioEngineering. ICBM data are disseminated by the Laboratory of Neuro Imaging at the University of Southern California.

The NACC database is funded by NIA/NIH Grant U24 AG072122. NACC data are contributed by the NIA-funded ADRCs: P30 AG062429 (PI James Brewer, MD, PhD), P30 AG066468 (PI Oscar Lopez, MD), P30 AG062421 (PI Bradley Hyman, MD, PhD), P30 AG066509 (PI Thomas Grabowski, MD), P30 AG066514 (PI Mary Sano, PhD), P30 AG066530 (PI Helena Chui, MD), P30 AG066507 (PI Marilyn Albert, PhD), P30 AG066444 (PI John Morris, MD), P30 AG066518 (PI Jeffrey Kaye, MD), P30 AG066512 (PI Thomas Wisniewski, MD), P30 AG066462 (PI Scott Small, MD), P30 AG072979 (PI David Wolk, MD), P30 AG072972 (PI Charles DeCarli, MD), P30 AG072976 (PI Andrew Saykin, PsyD), P30 AG072975 (PI Julie Schneider, MD), P30 AG072978 (PI Neil Kowall, MD), P30 AG072977 (PI Robert Vassar, PhD), P30 AG066519 (PI Frank LaFerla, PhD), P30 AG062677 (PI Ronald Petersen, MD, PhD), P30 AG079280 (PI Eric Reiman, MD), P30 AG062422 (PI Gil Rabinovici, MD), P30 AG066511 (PI Allan Levey, MD, PhD), P30 AG072946 (PI Linda Van Eldik, PhD), P30 AG062715 (PI Sanjay Asthana, MD, FRCP), P30 AG072973 (PI Russell Swerdlow, MD), P30 AG066506 (PI Todd Golde, MD, PhD), P30 AG066508 (PI Stephen Strittmatter, MD, PhD), P30 AG066515 (PI Victor Henderson, MD, MS), P30 AG072947 (PI Suzanne Craft, PhD), P30 AG072931 (PI Henry Paulson, MD, PhD), P30 AG066546 (PI Sudha Seshadri, MD), P20 AG068024 (PI Erik Roberson, MD, PhD), P20 AG068053 (PI Justin Miller, PhD), P20 AG068077 (PI Gary Rosenberg, MD), P20 AG068082 (PI Angela Jefferson, PhD), P30 AG072958 (PI Heather Whitson, MD), P30 AG072959 (PI James Leverenz, MD).

Data was provided by OASIS-3. Longitudinal Multimodal Neuroimaging: Principal Investigators: T. Benzinger, D. Marcus, J. Morris; NIH P30 AG066444, P50 AG00561, P30 NS09857781, P01 AG026276, P01 AG003991, R01 AG043434, UL1 TR000448, R01 EB009352. AV-45 doses were provided by Avid Radiopharmaceuticals, a wholly owned subsidiary of Eli Lilly.

Data contributed from ROS/MAP/MARS was supported by NIA R01AG017917, P30AG10161, P30AG072975, R01AG022018, R01AG056405, UH2NS100599, UH3NS100599, R01AG064233, R01AG15819 and R01AG067482, and the Illinois Department of Public Health (Alzheimer's Disease Research Fund). Data can be accessed at www.radc.rush.edu.

The data contributed from the Wisconsin Registry for Alzheimer's Prevention was supported by NIA AG021155, AG0271761, AG037639, and AG054047.

This work was supported by the Alzheimer's Disease Sequencing Project Phenotype Harmonization Consortium (ADSP-PHC) that is funded by NIA (U24 AG074855, U01 AG068057 and R01 AG059716).

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

## Appendix A. Additional Dataset Information

**ADNI**: Data used in the preparation of this article were obtained from the Alzheimer's Disease Neuroimaging Initiative (ADNI) database (adni.loni.usc.edu). The ADNI was launched in 2003 as a public-private partnership, led by Principal Investigator Michael W. Weiner, MD. The primary goal of ADNI has been to test whether serial magnetic resonance imaging (MRI), positron emission tomography (PET), other biological markers, and clinical and neuropsychological assessment can be combined to measure the progression of mild cognitive impairment (MCI) and early Alzheimer's disease (AD). Information about the dMRI sequence used in the present study is provided below. For further details, please visit https://adni.loni.usc.edu/.

**NACC**: The NACC cohort began in 1999 and is comprised of dozens of Alzheimer's Disease Research Centers that collect multimodal AD data[a]. The overall intention of the NACC cohort is to collate a large database of standardized clinical/neuropathological data[b,c,d,e]. [a]Beekly DL, Ramos EM, van Belle G, et al. The national Alzheimer's coordinating center (NACC) database: an Alzheimer disease database. Alzheimer Disease & Associated Disorders. 2004;18(4):270-277. [b]Beekly DL, Ramos EM, Lee WW, et al. The National Alzheimer's Coordinating Center (NACC) database: the uniform data set. Alzheimer Disease & Associated Disorders. 2007;21(3):249-258.48. [c]Besser LM, Kukull WA, Teylan MA, et al. The revised National Alzheimer's Coordinating Center's Neuropathology Form—available data and new analyses. Journal of Neuropathology & Experimental Neurology. 2018;77(8):717-726.49. [d]Weintraub S, Besser L, Dodge HH, et al. Version 3 of the Alzheimer Disease Centers' neuropsychological test battery in the Uniform Data Set (UDS). Alzheimer disease and associated disorders. 2018;32(1):10.50. [e]Weintraub S, Salmon D, Mercaldo N, et al. The Alzheimer's disease centers' uniform data set (UDS): The neuropsychological test battery. Alzheimer disease and associated disorders. 2009;23(2):91.

## Appendix B.  Visualization of Generated Samples

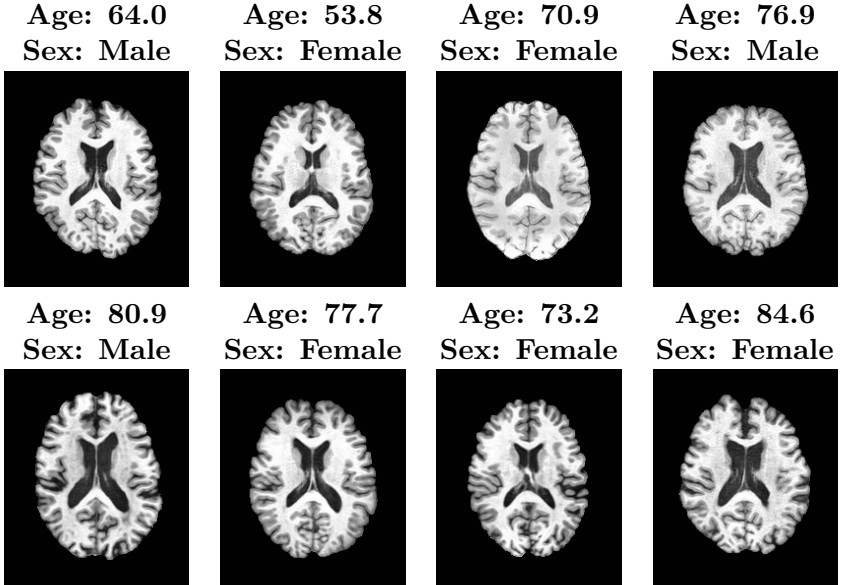

Figure B.1: Unconditional generated samples by MetaVoxel. All images, age, and sex shown are decoded from the latent space before visualization.

**Input**  **Inpainting with Left Half of the Image Fixed**

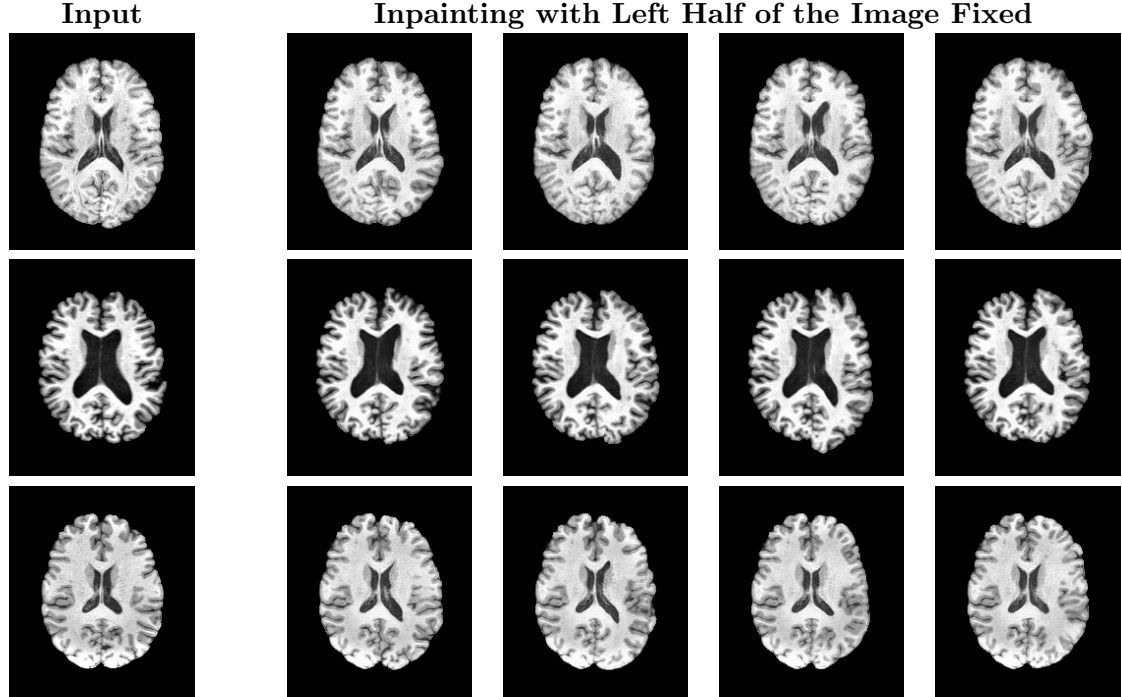

Figure B.2: Image inpainting with MetaVoxel. Pixels in the left half of the image are repeatedly overwritten with their known values from the noised input during the denoising process, while the remaining pixels are freely generated. Each row shows a different example. MetaVoxel produces visually coherent completions that smoothly blend with the fixed portion of the image.

