# OpenReview forum: "MetaVoxel: Joint Diffusion Modeling of Imaging and Clinical Metadata"
_MIDL.io/2026/Conference — MIDL 2026 Poster_

### Official Review · Reviewer_BVeR · 2026-01-08

**Confidence:** 3
**Preliminary Rating:** 3
**Final Rating:** 3

**Summary:**

MetaVoxel proposes a joint multi-modal diffusion model that learns a single generative process over images, and clinical information using diffusion for continuous and discrete diffusion components. The core idea of the paper is about unifying multiple clinical variables under one generative model that can flexibly condition on whatever information is available or wanted.

**Strengths:**

- The novelty of this work is about the modeling of joint distributions p(I, A, S) rather than a traditional conditional method for diffusion based models.
- Zero-shot conditional mechanism allows the users can perform multiple tasks from one trained model without re-training the model
- Unlike image diffusion models, MetaVoxel allows to generate mixed data types from on single model
- Theoretically, based on strong ground, and proposed sound method for by referencing solid sources

**Weaknesses:**

Generally I think experiment is the weakness of this paper. They could have done more through experiments to showcase the proposed model.
- This paper has limited data type. I would like to see the author try this method with different modality of images or even more clinical meta-data if that is possible. Since this method is not limited to certain data types, trying other image modalities would be very interesting.
- The author only provided FID for image, MAE for age, and ACC for sex prediction. I would like to see more types of metrics especially for images. Or maybe some clinical evaluation if that is possible
- Specifically, I also want to see LDM baseline models that are conditioned on sex and age in traditional conditioning methods. I think it is not a fair comparison to compare LDM only trained with T1 image with MetaVoxel that trained with image, sex, and age.

**Detailed Comments:**

- This is a minor suggestion, but it looks like the actual figure and descriptions do not fully match. It would be nice if this can be more clearly explained, or replaced by another figure.
- I also like to see visualization of different models (ex) LDM) results and MetaVoxel side by side so we can compare them visually too.
- I would like to see a little more experimental details. General structure is described, but little lack of details.
- Also, it would be great to see more details of the data. For example I would like to see age/sex range/distribution in the dataset.

**Justification Of Final Rating:**

Great thanks to authors for hard work to address reviewer's concern. However, I will keep my rating same. I think this paper has some fundamental weakness in data formatting and it application. It is novel idea, but might need better experimental performance.

**Justification Of The Preliminary Rating:**

MetaVoxel is a novel idea that redesigned the diffusion pipeline. This suggests a foundational model that can handle multi-task with flexible conditioning based on the user's need. Also the theoretical background is sound, and the whole method pipeline is clearly explained throughout the paper. However, I would like to see more detailed experiments that can clearly demonstrate the novelty and superior aspects of the MetaVoxel over traditional diffusion based models.

**Questions To Address In The Rebuttal:**

I would like to see authors address some of the issues I mentioned on weakness and detailed comments.

---

> ### Author Response · Authors · 2026-01-20
>
> We thank the reviewer for the comments. Below, we address each point, outlining planned revisions, and requesting clarification where additional specificity would help us respond precisely.
>
> ⸻
>
> Comment 1: Limited scope of data types and modalities
>
> We appreciate this suggestion and agree that extending MetaVoxel to additional imaging modalities and richer clinical metadata is a promising direction. The current experimental design is intended as a proof of concept for joint diffusion modeling, demonstrating, for the first time, how learning a joint distribution can enable multiple downstream tasks within a single diffusion framework. T1-weighted MRI, age, and sex were selected because they span three distinct data types and are widely available across datasets. We view broader experiments as a natural extension of this work and plan to explore them in future studies.
>
> ⸻
>
> Comment 2: Limited evaluation metrics
>
> We will add Maximum Mean Discrepancy (MMD) as an additional evaluation metric. Metrics such as SSIM or PSNR, which are commonly reported in prior work, are designed for settings where a ground-truth image is available for direct comparison and are therefore not applicable in our unpaired generative setting. If the reviewer has specific additional metric suggestions, we would be happy to incorporate them.
> ⸻
>
> Comment 3: Lack of conditional LDM baselines using age and sex
>
> We are currently training the suggested conditional LDM baselines and will include these results in the revised manuscript.
>
> ⸻
>
> Comment 4: Mismatch between figures and descriptions
>
> We have carefully re-examined the manuscript and were unable to identify a clear mismatch between the figures and their descriptions. To ensure that we address this concern accurately, we kindly ask the reviewer to specify which figure(s) and which aspect of the description appears inconsistent. We are fully willing to revise or replace figures if an issue is identified.
>
> ⸻
>
> Comment 5: Lack of side-by-side visual comparison between models
>
> We will include visual comparisons between MetaVoxel and baseline models in the revised manuscript.
>
> ⸻
>
> Comment 6: Insufficient experimental details
>
> We aimed to provide a concise yet complete description of the experimental setup. To ensure that any revisions are targeted and meaningful, we respectfully ask the reviewer to clarify which specific experimental details are missing or unclear. We would be happy to expand the corresponding sections accordingly.
>
> ⸻
>
> Comment 7: Dataset statistics and demographic distributions
>
> We will add a concise summary of dataset demographics (e.g., age range and sex distribution) in the appendix.

---

> > ### Author Response · Authors · 2026-01-23
> > **Updates on previous comment**
> >
> > Update on Comment 3: Lack of conditional LDM baselines using age and sex
> >
> > We include additional experiments in Appendix D comparing MetaVoxel to a class-conditional LDM in a conditional image generation setting. Specifically, we report FID and MMD stratified by age and sex groups, demonstrating that MetaVoxel remains competitive and shows its flexibility in adapt to new task without any modification to the model or weights.
> >
> > We further clarify that the experiments in Section 4.1 deliberately focus on unconditional sampling from the joint distribution. This setting is necessary to demonstrate MetaVoxel’s ability to represent and sample from the full joint distribution over imaging and clinical variables. Under this objective, an unconditional LDM constitutes the most direct and appropriate baseline. In contrast, a class-conditional LDM requires conditional inputs at test time and therefore addresses a different problem setting.
> >
> > ⸻
> >
> > Update on Comment 5: Lack of side-by-side visual comparison between models
> >
> > We have updated the figures in Appendix B to include visual comparisons between MetaVoxel and baseline models. We note that each model generate samples independently from the data distribution, no pixel-level correspondence exists and direct voxel-wise comparison is not meaningful.
> >
> > ⸻
> >
> > Update on Comment 7: Dataset statistics and demographic distributions
> >
> > We have added a summary of dataset statistics, including age range and sex distribution, in Appendix C.

---

### Official Review · Reviewer_5ovz · 2026-01-10

**Confidence:** 5
**Preliminary Rating:** 2
**Final Rating:** 2

**Summary:**

This paper proposed a model called MetaVoxel, a diffusion network to model multiple (all) available variables from brain MRI data.
The model is trained on 10000 MRI data and tested on image generation, age and sex prediction.

**********************************************************************************************

**Strengths:**

The idea of a single model generally applicable to a variety of tasks, vs specific classifiers for each condition, is an interesting one.
The idea of using a diffusion model seems reasonable.

***********************************************

**Weaknesses:**

Unfortunately, the accuracy of prediction is quite a bit below recent published methods.

For example, Table 1 shows results. Sex prediction accuracies of (0.788, 0.884) are reported for densenet and transformer architectures, these are compared favorably against (0.815, 0.855) accuracies for the MetaVoxel variants.

However other approaches[a][b] achieve accuracy of (0.92-0.95) for sex classification of rigidly registered brain MRI data. So the accuracy of the proposed MetaVoxel seems quite inferior to actual results in the literature on sex classification. Furthermore this literature is missing from the paper discussion.

[a] Ebel, Matthis, et al. "Classifying sex with volume-matched brain MRI." Neuroimage: Reports 3.3 (2023): 100181.

[b] Toews, M., et al.  (2025). Representative scale-invariant characteristics of male and female brains in magnetic resonance images. NeuroImage: Reports, 5(3), 100267.

**Detailed Comments:**

See comments above. The model and idea seem good. The results are lacking.

**Justification Of Final Rating:**

The rebuttal refuses to cite or consider relevant work on sex classification. The results here should be better considering that subjects have not been normalized for size, however they are unconvincingly low, calling into question the usefulness of this model.

Furthermore, the SIFT has been used to classifying any image data (e.g. Lung CT, brain MRI, cardiac ultrasound) with arbitrary labels and no training, preprocessing, the authors should consider this while trying to reinvent similar capability.
*************

**Justification Of The Preliminary Rating:**

The model seems interesting, a foundational model for brain MRI. However, the classification accuracy is quite a bit lower than other reported methods.
************************************************************

**Questions To Address In The Rebuttal:**

The above comments

---

> ### Author Response · Authors · 2026-01-20
> **Sex prediction accuracy relative to prior literature**
>
> We appreciate the effort in raising concerns regarding the numerical comparison with prior sex classification studies. We have carefully reviewed the works mentioned by the reviewer.  Below, we clarify why the comparisons, while intuitive at first glance, do not reflect equivalent problem settings.
>
> 1. The referenced works address sex classification under substantially different input definitions.
>
> Ebel et al. (2023) operates on voxel-wise gray-matter volume (GMV) maps generated through a dedicated pipeline involving tissue segmentation, spatial normalization, modulation, and explicit volume matching.
>
> Toews et al. (2025) relies on hand-crafted 3D SIFT keypoints, extracted after standardized HCP preprocessing and registration, followed by feature matching.
>
> In contrast, MetaVoxel operates directly on skull-stripped T1-weighted MRI intensities, without external knowledge such as segmentation labels, hand-engineered features. As a result, MetaVoxel solves a less constrained problem. Accuracy values across these settings therefore reflect different task definitions rather than differences in model quality.
>
> 2. Their evaluation settings are substantially more controlled than our study.
>
> Toews et al. (2025) evaluate on 422 subjects from a single site (HCP), restricted to young adults aged 22–36, with tightly controlled acquisition protocols and preprocessing.
>
> Ebel et al. (2023) rely on cohort-matched and volume-matched subsets, primarily drawn from controlled cohorts (SHIP and HCP), with explicit demographic and anatomical balancing.
>
> In contrast, MetaVoxel is trained and evaluated on over 10,000 subjects drawn from 9 datasets, spanning a wide adult lifespan, multiple scanners, sites, and acquisition protocols. The comparison therefore favors the cited works while penalizing MetaVoxel for operating in a multi-site, heterogeneous, and clinically realistic regime.
>
> We also note that the reviewer referenced works report a wide range of sex classification accuracies. Ebel et al. (2023) explicitly summarize prior results spanning approximately 69%–96% and report their own results ranging from roughly 76%–95% across configurations. These ranges overlap directly with the sex prediction accuracies observed for MetaVoxel and its baselines
>
> Finally, we emphasize that achieving state-of-the-art performance on sex classification is not the primary objective of MetaVoxel. Sex prediction is included as a representative categorical variable to demonstrate joint diffusion modeling and zero-shot conditional inference within a single unified framework capable of supporting a broad range of downstream tasks. As stated in the introduction, our goal is not to compete with highly specialized discriminative pipelines optimized for a single endpoint, but rather to show that a single joint generative model can perform image generation, regression, and classification without task-specific retraining, while additionally enabling flexible conditioning and sample-based uncertainty estimation. Our presented experiments support this claim.
>
> We hope this clarification helps contextualize the reported results and addresses the reviewer’s concerns regarding the comparison.

---

> > ### Comment · Reviewer_5ovz · 2026-01-29
> > **refuse to cite directly related research**
> >
> > The authors discard existing research on the same task with strawman arguments. Eg the SIFT approach requires no training, no preprocessing, applies to any data, achieves similar results here.

---

> > > ### Author Response · Authors · 2026-01-29
> > > **Concern Regarding Reviewer Discussion**
> > >
> > > Dear Program Chair and Area Chair,
> > >
> > > We are writing to confidentially bring to your attention a concern regarding the ongoing discussion with Reviewer 5ovz, as it has shifted away from constructive technical exchange.
> > >
> > > Briefly, the original reviewer comment questioned the comparability of our reported results to prior work on sex prediction. In our response, we explained that direct numerical comparison with the cited studies is not meaningful due to substantial differences in input information and cohort structure, noted that the reported accuracy ranges in those works overlap with ours, and clarified that sex prediction in our study is included to illustrate joint diffusion modeling rather than as a task-specific state-of-the-art objective.
> > >
> > > The reviewer’s reply does not address the technical distinction we raised. Instead, it reframes our clarification as a dismissal of prior work and asserts that “the SIFT approach requires no training, no preprocessing, and applies to any data.” This statement inaccurately characterizes the cited method and does not engage with the methodological differences we described.
> > >
> > > We fully support MIDL’s discussion phase and appreciate its goal of enabling constructive technical exchange. However, the latest response shifts from technical evaluation to an accusation about our clarification and  misrepresentation of the cited literature. We therefore believe that further exchange is unlikely to improve the scientific assessment of the paper, and for this reason we have chosen not to continue the discussion thread.
> > >
> > > We submit this note solely for your awareness, as the current exchange may otherwise appear as an unresolved technical dispute or unresponsive on our side.
> > >
> > > Thank you for your time and consideration.

---

### Official Review · Reviewer_yBVw · 2026-01-12

**Confidence:** 4
**Preliminary Rating:** 2
**Final Rating:** 2

**Summary:**

The authors propose MetaVoxel, a multimodal diffusion framework that learns the joint distribution of 3D T1-weighted MRI images and clinical metadata (Age and Sex). Unlike standard task-specific models, MetaVoxel learns the joint distribution (Image, Age, Sex), allowing for flexible inference tasks (generation, regression, classification) using a single model.

**Strengths:**

1. The MetaVoxel model the joint distribution p(I, A, S) is a conceptually novel approach. It simplifies the deployment of medical AI by enabling multiple downstream tasks (generation, regression, classification) using a single trained model, avoiding the need for separate task-specific models.
2. The utilization of a large, multi-site dataset of over 10,000 subjects demonstrates the scalability of the method. The experiments demonstrate the proposed framework is effective.

**Weaknesses:**

1. The Introduction explicitly motivates this work with clinical questions such as "Does this individual show signs of a particular disease?" and "How would the imaging appear if disease status were different?" However, Section 4 states that "Only cognitively unimpaired individuals were included" in the dataset. By training exclusively on healthy subjects and excluding disease labels from the joint distribution, the current implementation of MetaVoxel is structurally incapable of addressing the primary diagnosis and disease generation tasks.
2. In section 2 Background and Related Works, it uses excessive space for “Discrete diffusion models” and “Unconditioned DDPMs”, which could be condensed to single sentences with citations. Conversely, the section on "Conditional DDPMs" lacks references to foundational basic class-conditional diffusion models.
3. In section 4 Datasets, the data was split into approximately 89% of subjects for training, 1% for validation, and 10% for testing. The validation set of only 108 subjects is dangerously small and creates a high risk of overfitting. It cannot reliably represent the distribution of the 9,000+ training cases.
4. In section 4.1, the authors state “we generated 100 synthetic T1w scans and quantified sample quality using the FID.” The disparity between the reference set size N=968 and the generated set size N=100 introduces a sample-size bias in the FID calculation, as FID is known to be sensitive to N.
5. The authors utilize a slice-based 2D FID metric to evaluate a 3D volumetric model. While the multi-plane approach (MAISI) partially addresses spatial consistency, could the authors discuss why explicit 3D metrics were not employed.
6. The paper compares MetaVoxel against an unconditional LDM. The most relevant baseline is missing: a standard Class-Conditional Diffusion Model (e.g., a standard LDM conditioned on Age and Sex via cross-attention).

**Detailed Comments:**

1. In Section 3.1 Encoding, the authors state “We implement this VAE by extending the 2D architecture used in LDM (Rombach et al., 2022) to full 3D, and we train it using the same loss formulation employed in MAISI (Guo et al., 2025).” The description of the VAE relies heavily on citations. While citing is appropriate, a brief summary of the specific 3D architecture and the loss components is also needed to assist readers without reading more papers.
2. The manuscript mentions using a VAE for dimensionality reduction. It is unclear if this VAE was: (a) Pre-trained on a separate dataset and frozen? (b) Trained on this dataset and frozen? (c) Trained end-to-end with the diffusion model?
3. In section 3.1 Joint Diffusion, the paper mentions concatenating scalar variables (Age/Sex) with the high-dimensional image latent space. How exactly is a scalar concatenated with a 3D tensor? The text mentions "input-channel concatenation," which implies broadcasting, but this increases memory usage significantly.

**Justification Of Final Rating:**

The primary purpose of medical imaging research is to assist in clinical diagnosis. Therefore, a model trained exclusively on healthy subjects lacks clinical validity and fails to demonstrate any practical utility for the medical tasks it claims to address.The 1% validation split falls significantly below the universally recognized standards in large-scale machine learning. Given these fundamental flaws in both clinical relevance and methodological rigor, I maintain my original score.

**Justification Of The Preliminary Rating:**

My preliminary rating is Weak Reject. While the proposed MetaVoxel framework offers an novel solution for joint multimodal modeling, the current experimental design has critical weaknesses that undermine the validity of the results.
The most significant flaw is the use of a 1% validation split (108 subjects). Given the high heterogeneity of the multi-site data, this sample size is statistically insufficient for robust hyperparameter tuning and early stopping, creating a high risk of overfitting to a non-representative subset. Furthermore, a critical contradiction exists between the study's motivation and its methodology, which excludes all disease cases from training. This structural limitation renders the model incapable of performing the tasks it aims to solve. Finally, the absence of a standard class-conditional baseline and the presence of sample-size bias in the FID evaluation further weaken the empirical evidence. I am open to reconsidering if the authors can resolve these concerns in the rebuttal.

**Questions To Address In The Rebuttal:**

1. Validation Split: Can you demonstrate that your model selection (based on only 108 validation subjects) is stable and generalizes to the test set? Specifically, can you provide results from a run with a larger validation split (e.g., 10%).
2. FID Bias: Can you provide the balanced FID scores (N=968 or 100) to compare the performance.
3. Baseline: Can you provide the comparison with Class-Conditional LDM.
4. Disease Modeling: How does the model address the clinical motivation of "disease diagnosis" when it was trained exclusively on cognitively unimpaired subjects?

---

> ### Author Response · Authors · 2026-01-23
>
> We thank the reviewer for the detailed and thoughtful feedback. Below, we address the main concerns point by point and clarify several aspects of the experimental design and methodological choices.
>
> ⸻
>
> Validation split and overfitting concerns
>
> We thank the reviewer for raising concerns about the validation set size. We believe that the comment involve two notions of overfitting:
> 	1.	Overfitting to the training set, which would manifest as decreasing training loss alongside increasing validation loss.
> 	2.	Overfitting to the validation set, where model selection favors a “lucky” checkpoint that performs well on validation but poorly on test data.
>
> While we acknowledge that the statement “dangerously small” is subjective, we agree that empirical evidence is appropriate. Rerunning the full set of experiments, including all baselines and diffusion models, with an alternative data split is infeasible within the rebuttal period due to computational constraints. Instead, we address this concern by providing additional experiments using fast-to-train baselines in our experiments (3D DenseNet and 3D ViT). These models use the same data split and would be equally susceptible to overfitting if such an issue were present
>
> Using the original split, we report training, validation, and test losses across epochs (https://github.com/yihao6/midl-2026-rebuttal/blob/main/loss_comparisons/loss_comparison.png). These curves show that:
> (i) validation loss does not increase as training proceeds, indicating no overfitting to the training set; and
> (ii) test loss remains stable across a wide range of epochs, indicating no overfitting to the validation set.
>
> Moreover, with nearly 10,000 3D volumes for training, checkpoint-level overfitting is unlikely in practice, particularly since we do not perform exhaustive checkpoint selection at every weight update. Finally, the purpose of a validation set is not to represent the full training distribution, but to provide an unbiased draw from the same distribution as the test set, which is satisfied in our experiment.
>
> ⸻
>
> FID sample-size bias
>
> We apologize for the lack of clarity in the original text. Following the MAISI implementation (https://github.com/Project-MONAI/tutorials/blob/main/generation/maisi/scripts/compute_fid_2-5d_ct.py), FID was computed using balanced sample sizes, with 100 generated samples compared against 100 randomly selected test samples. We have updated the manuscript to make this explicit.
>
> ⸻
>
> Clinical motivation and disease modeling
>
> The reviewer notes a mismatch between the clinical questions posed in the Introduction and the use of cognitively unimpaired subjects in the experiments. We clarify that the opening paragraph of the Introduction is intended to motivate joint modeling of imaging and clinical metadata by illustrating the multidimensional nature of clinical reasoning, rather than to enumerate tasks explicitly addressed in the experiments.
>
> As emphasized by the title and the stated contributions, this work focuses on joint diffusion modeling rather than disease-specific diagnosis or progression modeling. The MetaVoxel framework itself imposes no restriction on the inclusion of disease variables; disease labels or biomarkers can be incorporated as additional variables in the joint distribution without architectural changes. In this initial study, we selected T1-weighted MRI, age, and sex because they are widely available across datasets and allow us to demonstrate joint modeling across heterogeneous data types (high-dimensional continuous images, continuous scalars, and categorical variables). Given the absence of closely related prior work on joint diffusion over imaging and clinical metadata, our primary goal is to establish feasibility and characterize behavior across generation, regression, and classification tasks. Extending the framework to disease modeling is an important and natural next step that we plan to explore in future work.

---

> > ### Author Response · Authors · 2026-01-23
> >
> > Slice-based FID and lack of explicit 3D metrics
> >
> > Our evaluation focuses on unconditional 3D image generation, which requires comparing distributions. 3D FID would require training a custom 3D feature extractor, which introduces additional confounds and limits reproducibility. We therefore adopt the slice-based strategy implemented in MAISI, which leverages pretrained networks.
> > Metrics such as SSIM, PSNR, or MSE require voxel-wise correspondence and are appropriate for reconstruction or translation tasks where a single ground truth image is available, but are not applicable for comparing distributions of images. In the revised manuscript, we additionally include Maximum Mean Discrepancy (MMD) with a Gaussian RBF kernel as an additional metric.
> >
> > ⸻
> >
> > Baseline comparison with class-conditional diffusion models
> >
> > The experiment in section 4.1 shows unconditional sampling of MetaVoxel, which is necessary to demonstrate its ability to represent and sample from the full joint distribution. Thus, unconditional LDM is a direct comparison method. A class-conditional LDM requires conditional inputs at test time and thus focus on a different problem setting.
> > We directly address the reviewer’s request by including additional experiments in the appendix D, comparing MetaVoxel to a class-conditional LDM in a conditional image generation setting. We report FID and MMD stratified by age and sex groups, demonstrating that MetaVoxel remains competitive and shows its flexibility in adapt to new task without any modification to the model or weights.
> >
> > ⸻
> >
> > VAE training protocol
> >
> > We clarify that the VAE is trained on the same dataset prior to diffusion training and is subsequently frozen, following standard LDM practice. We have added an explicit statement to Section 3.1 to avoid ambiguity.
> >
> > ⸻
> >
> > Scalar–tensor concatenation and memory usage
> >
> > Scalar variables (age and sex) are broadcast along spatial dimensions and concatenated at the input-channel level, consistent with common practice in conditional diffusion models. The additional memory overhead is negligible relative to the dominant cost of high-dimensional feature maps in the 3D U-Net. For example, a single convolutional layer operates on tens of feature channels per voxel, far exceeding the cost of broadcasting few scalar channels at the input.

---

### Author Rebuttal · Authors · 2026-01-23

**Rebuttal:**

We thank the reviewers for the time and for the constructive comments provided on our manuscript. We have carefully considered all the reviewers’ suggestions and have revised the paper accordingly. The revised manuscript is provided in attachment. For ease of review, all modifications in the revised manuscript have been highlighted with magenta color and change bars.

**Supporting Material:**

/attachment/16a9b9cd07d51d006eb8d8545afe2e1585668c1d.pdf

---

### Meta-Review · Area_Chair_AC42 · 2026-02-03

**Recommendation:** Reject
**Confidence:** 4

**Metareview:**

The AC recognised that the method shows nice novelty in terms of methods, but the reviewers raised consistent concerns about its validation and clinical validity. Given the reviewers rating for the fundamental flaws in both clinical relevance and validation rigor, the current version may not be sufficiently good for MIDL. By taking into account the reviewers' suggestions, it would be a solid and impactful work in this community.

---

### Decision · Program_Chairs · 2026-02-13

**Decision:**

Accept (Poster)

**Comment:**

Although the meta-review recommendation was negative, the Area Chair acknowledged the strong methodological novelty and noted that, with revisions, this work could be solid and impactful. After extensive discussion, we recognize concerns about validation and clinical positioning, but we believe these are addressable. Given the originality of the idea and its potential long-term impact on multimodal medical AI, we lean to accept this paper to MIDL.